# Application of Response Surface Methodology for Ethanol Conversion into Hydrocarbons Using ZSM-5 Zeolites

**José Faustino Souza de Carvalho Filho [1,2,]***, **Marcelo Maciel Pereira [1]**,
**Donato Alexandre Gomes Aranda [2]**, **João Monnerat Araujo Ribeiro de Almeida [2]**,
**Eduardo Falabella Sousa-Aguiar [2]** and **Pedro Nothaft Romano [3]**

[1]  Instituto de Química, Universidade Federal do Rio de Janeiro, Rio de Janeiro 21941-909, Brazil
[2]  Escola de Química, Universidade Federal do Rio de Janeiro, Rio de Janeiro 21941-909, Brazil
[3]  Departamento de Nanotecnologia, Universidade Federal do Rio de Janeiro, Campus Duque de Caxias, Rio de Janeiro 25265-970, Brazil
*  Correspondence: faustinocarvalho@gmail.com; Tel.: +55-21-971808257

**Abstract:** The ethanol conversion into hydrocarbons (light olefins and aromatics) using alkali-treated HZSM-5 with different $SiO_2/Al_2O_3$ ratios (23, 38, and 53) zeolites was evaluated. The desilicated SAR 38 zeolite exhibited significant growth on the external surface area (61–212 $m^2$/g) and the mesopore volume (0.07–0.37 $cm^3$/g) without significate reduction on XRD crystallinity (93%). All catalysts were active on the ethanol conversion into hydrocarbons. At the same set of variables, the alkali-treated HZSM-5 zeolites showed a better conversion and a high selectivity to $C_4$–$C_9$ hydrocarbons when compared to the parent microporous zeolites. Only the parent HZSM-5 zeolite (SAR 53) was chosen for the statistical study using the standard response surface methodology in combination with the central composite design. It was found that maximum BTEX (benzene, toluene, ethylbenzene, and xylenes) and minimum ethylene production were reached for the following conditions: temperature 450 °C, pressure 20 bar, and WHSV (weight hourly space velocity) 5 $h^{-1}$.

**Keywords:** HZSM-5; ethanol dehydration; optimization; desilication; response surface methodology

## 1. Introduction

Currently, ethylene is mainly produced by steam cracking of crude oil derivatives. However, ethanol can also be used as raw material for ethylene production. In recent years, catalytic conversion of ethanol into hydrocarbons has attracted considerable attention, since that alcohol can be used as a green raw material capable of providing an alternative route to produce a great range of valuable chemicals for the petrochemical industry [1–4]. In this context, the obtaining of light olefins and BTEX aromatics (benzene, toluene, ethylbenzene and xylenes) from alcohols have attracted notable attention, since the alcohol-to-jet (ATJ) technology can provide a renewable route to obtain kerosene from light alcohols, such as ethanol and butanol [5,6].

Due to their characteristics, zeolites are the most promising catalysts for this kind of reaction, among them, the HZSM-5 zeolite is the most effective catalyst for conversion of ethanol into $C_3$–$C_8$ hydrocarbons and BTEX [7–9]. Sousa et al. [10] investigated the ethanol conversion into olefins and aromatics using HZSM-5 zeolites. According to the authors, the temperature programmable surface reaction (TPSR) showed that at temperatures between 150 °C and 250 °C ethanol was converted into ethylene by means of two subsequent dehydration reactions. Intermolecular dehydration led to diethyl ether production, and its formation reached the maximum at 200 °C. Above that value, temperature increase favored ethylene formation. Furthermore, temperatures higher than 300 °C

favored the direct ethanol intramolecular dehydration into ethylene and light olefins formation via ethylene oligomerization.

The catalyst acidity plays a key role in alcohol dehydration over ZSM-5 zeolites, excessive acidity generally deactivates the catalyst due to coke formation [8]. Oliveira et al. [11] studied the ethanol dehydration into diethyl ether over Cu-Fe-ZSM-5 catalysts. According to the authors, the Cu and Fe impregnated zeolites showed significant changes on their crystallinity, surface acidity, and catalytic properties in comparison to the parent zeolite. The impregnation of Cu to the zeolite promoted a decrease in its acid strength but did not affect the ZSM-5 crystallinity. That modification led to a decrease on the ethanol conversion and promote a significant increase in diethyl ether selectivity. The impregnation of Fe did not change the ZSM-5 acidity but did promote a decrease in its crystallinity.

Despite the fact that zeolites are widely employed as catalysts in many reactions, one of the greatest inconveniences in its use concerns to the diffusional limitations [12–15]. Molecular mobility plays an important role in long-chain hydrocarbons synthesis, slow diffusion or long residence time can favor the formation of bulky intermediates that promote deactivation by pore blocking mechanisms [15]. The synthesis of hierarchical zeolites is a common approach to attenuate the diffusional problems. Among the strategies taken to synthesize hierarchically-structured zeolites, one of the most reported in the literature is the obtaining of mesoporous zeolites. "Top-down" (post-synthetic modification) and "bottom-up" (primary synthesis using soft or hard templates) are the main groups where the different synthesis methods are placed. The top-down technique involves the removal of one of the two main components of zeolites, the desilication process leads to mesopore creation by removing silicon from the zeolite's structure, and the dealumination process, the opposite [15–17].

Xin et al. [18] studied the catalytic dehydration of ethanol over post-threated HZSM-5 zeolites. According to the authors, they turned a microporous ZSM-5 zeolite into a hierarchical pore structure solid by means of three top-down techniques: desilication with sodium hydroxide, dealumination with oxalic acid, and both of them in a sequential way. At 200 °C and ambient pressure, diethyl ether and ethylene were the major reaction products, and the $NH_3$-TPD data indicated that weak acid sites facilitated the ethylene production. Furthermore, Xin's group's theoretical calculation for the reaction pathways for diethyl ether and ethylene formation from ethanol indicated that both activation energies and natural charges of the transition states endorsed that the selectivity for ethylene tended to grow with reducing of Brønsted acidity. Textural properties data obtained from nitrogen-physisorption of the post-treated zeolites indicated significant modifications on the external surface area and mesopore volume. However, the choice of the authors to work at 200 °C did not provide a good comprehension of how the textural modifications affected the reaction products diffusion, and all five catalysts showed stable ethanol conversion and ethylene selectivity with 12 h time-one-stream (TOS).

Temperature, pressure, ethanol concentration, and weight hourly space velocity (WHSV) have a significant effect on the product distribution and catalyst activity on ethanol conversion over ZSM-5 zeolite. Sousa et al. [10] also investigated the influence of reaction conditions on ethanol conversion into olefins and aromatics over acidic ZSM-5 zeolites. The results showed that, at 500 °C, partial pressure of ethanol equal to 0.12 atm, and WHSV equal to 6.5 $h^{-1}$, the formation of propene was favored. Furthermore, the liquid hydrocarbon fraction (majorly aromatics) was favored by the lowest WHSV value (0.65 $h^{-1}$). In addition to the reaction parameters, Ramasamy et al. [8] studied the effect of $SiO_2/Al_2O_3$ ratio (SAR) on the products distribution of ethanol conversion to hydrocarbons over HZSM-5 zeolites. The results revealed that HZSM-5 zeolites with a high SAR presented a fast deactivation and produced more unsaturated liquid compounds. Temperatures close to 300 °C tended to promote a fast catalyst deactivation, generating a minor liquid hydrocarbon fraction. On the other hand, temperatures near to 400 °C tended to increase the gaseous products fraction and reduce the liquid hydrocarbons range. In addition, high WHSV values led to a higher ethylene formation. Ramasamy's group's results further indicated that changes in product composition was related to declining in the zeolite acid strength by acid-site poisoning and zeolite pore blocking due to coke deposition along the walls and pore entrance.

A wealth of papers discussing how the reaction conditions affect the distribution of the products in one-pot ethanol conversion into hydrocarbons is available [19–22]. However, most of them are based on a classic approach where only one variable changes at a time. That kind of study does not efficiently explore the possibilities of interactions among reaction conditions. In the present study, the Central Composite Design (CCD) approach composed a set of experiments ranged temperature, space velocity, and pressure. Response surface methodology (RSM) was performed for process optimization. The empirical model of the optimum reaction parameters was calculated using suitable modeling techniques [23–25].

## 2. Results and Discussion

### 2.1. Catalyst Characterization

Figure 1 presents the XRD patterns of all four zeolites used as the catalyst in the ethanol conversion. As explained in more detail in the next section, the first step of the work was the synthesis of hierarchical zeolites and the use of these solids in the conversion of ethanol into hydrocarbons. The data obtained from XRD, XRF, and $N_2$ physisorption were used to (1) ensure that the zeolite structure was maintained after desilication; (2) to evaluate the chemical composition of the catalysts and calculate the SAR; and (3) to verify if the textural properties of alkali-treated zeolites are characteristic of mesoporous solids, respectively. As can be seen, the XRD patterns exhibited well-resolved diffraction peaks which matched to the ZSM-5 type structure, and a characteristic doublet at 6°–10° 2θ alongside with a triplet at 22°–25° 2θ was easily identified. The relative crystallinity calculated for the post-treated zeolites (Table 1) showed a slight decrease in its value, generally, and the hierarchical zeolite shows weaker diffraction peak intensity. This result indicates partial destruction of the parent zeolite's structure by the removal of framework silicon [12].

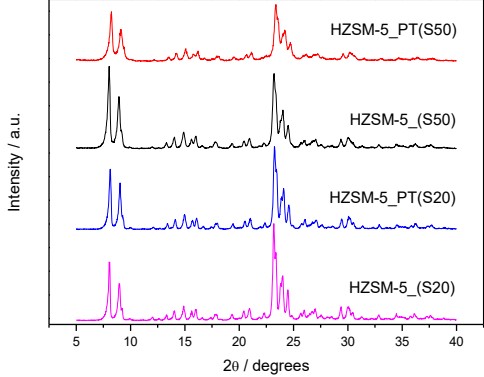

**Figure 1.** X-ray diffractograms of parent (HZSM-5_(S50) and HZSM-5_(S20)) and post-treated (HZSM-5_PT(S50) and HZSM-5_PT(S20)) zeolites.

The SAR values calculated from XRF data are presented in Table 1. There was no significant difference between the HZSM-5_(S20) and HZSM-5_PT(S20) zeolites. On the other hand, the difference between HZSM-5_(S50) and HZSM-5_PT(S50) indicated a higher silicon removal from the parent zeolite. The SAR value has a crucial influence on the creation of hierarchical porosity, the optimal SAR range for a silicon removing and well-controlled mesoporosity creation is 25–50 [26]. Lower SAR values inhibit the silicon removing by the protective effect of aluminum, and higher SAR values tend to promote excessive silicon extraction [26–28]. The adsorption-desorption isotherms for all four studied catalysts can be seen in Figure 2. Generally, the ZSM-5 zeolites (SAR 20 and 50) present a type I isotherm, which is characteristic for a microporous zeolite. However, hierarchical zeolites present a characteristic type IV isotherm, which displays a specific hysteresis loop at higher relative pressures related to the capillary condensation of $N_2$ in the mesoporous [12]. The two parent zeolites textural properties were affected by the alkali treatment, and the desilication had a smaller effect on the SAR 20

zeolite. The notable increase of the external area observed in the HZSM-5_PT(S50) catalyst was due to the mesopore formation. Additionally, the hierarchical zeolite mesopore volume (SAR 38) and pore diameter showed significant growth.

**Table 1.** Physicochemical and textural characterization of the parent and post-treated ZSM-5 zeolites.

| Catalyst | SAR [a] (molar) | Cristallinity [b] (%) | $S_{BET}$ [c] (m²/g) | $S_{meso}$ [d] (m²/g) | $V_{micro}$ [d] (cm³/g) | $V_{meso}$ [e] (cm³/g) |
|---|---|---|---|---|---|---|
| HZSM-5_PT(S50) | 38 | 93 | 491 | 212 | 0.10 | 0.37 |
| HZSM-5_(S50) | 53 | 100 | 419 | 61 | 0.18 | 0.07 |
| HZSM-5_PT(S20) | 23 | 99 | 352 | 42 | 0.11 | 0.04 |
| HZSM-5_(S20) | 23 | 100 | 337 | 12 | 0.16 | 0.02 |

[a] XRF; [b] XRD; [c] BET method; [d] t-Plot method; [e] BJH method (desorption branch).

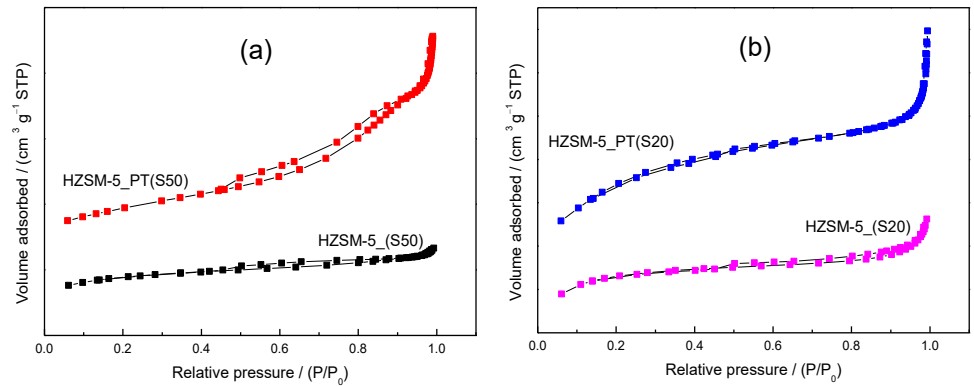

**Figure 2.** Nitrogen adsorption-desorption isotherms of parent and post-treated (**a**) SAR 50 zeolites and (**b**) SAR 20 zeolites.

## 2.2. Screening of Catalyst and Reaction Parameters

The influence of reaction parameters on the product distribution and ethanol conversion was initially verified, all four zeolites were tested at 360 °C, 15 bar, ~38 h⁻¹, and 3 h TOS. Figure 3 shows the product distribution of gaseous (a) and organic liquid (b) fractions, and ethanol conversion (c). In this specific case, the isoconversion approach was not used once this specific set of variables was evaluated. As can be observed, the SAR has a significant influence on the ethanol conversion, and zeolites with low SAR values (higher acidity) tend to be more ethylene selective. On the other hand, zeolites with high SAR values (lower acidity) tend to show higher ethanol conversion and generally promote the formation of heavier hydrocarbons. In this case, the HZSM-5_PT(S50) was the most stable catalyst on 3 h TOS, and that behavior was already expected since hierarchical zeolites promote a better mass transfer and higher external surface area facilitating the reagent/product diffusion and reducing the deactivation of the catalyst by pore blocking [15,22]. The organic liquid distribution was quite similar to all tested zeolites, however, there was no organic liquid product from the ZSM-5_(S20) catalytic test, and the amount of organic liquid product from the ZSM-5_PT(S50) test was approximately eight times greater than the ZSM-5_PT(S20). As mentioned above, the WHSV and the temperature have considerable effects on product distribution, and zeolites with high SAR and WHSV values generally tend to produce a considerable amount of ethylene and decrease the formation of heavier hydrocarbons. According to the screening results, considering the variability of the products obtained from HZSM-5_(S50) and the desire of testing the 15–55 h⁻¹ range for the WHSV parameter, that zeolite was chosen for the statistical study.

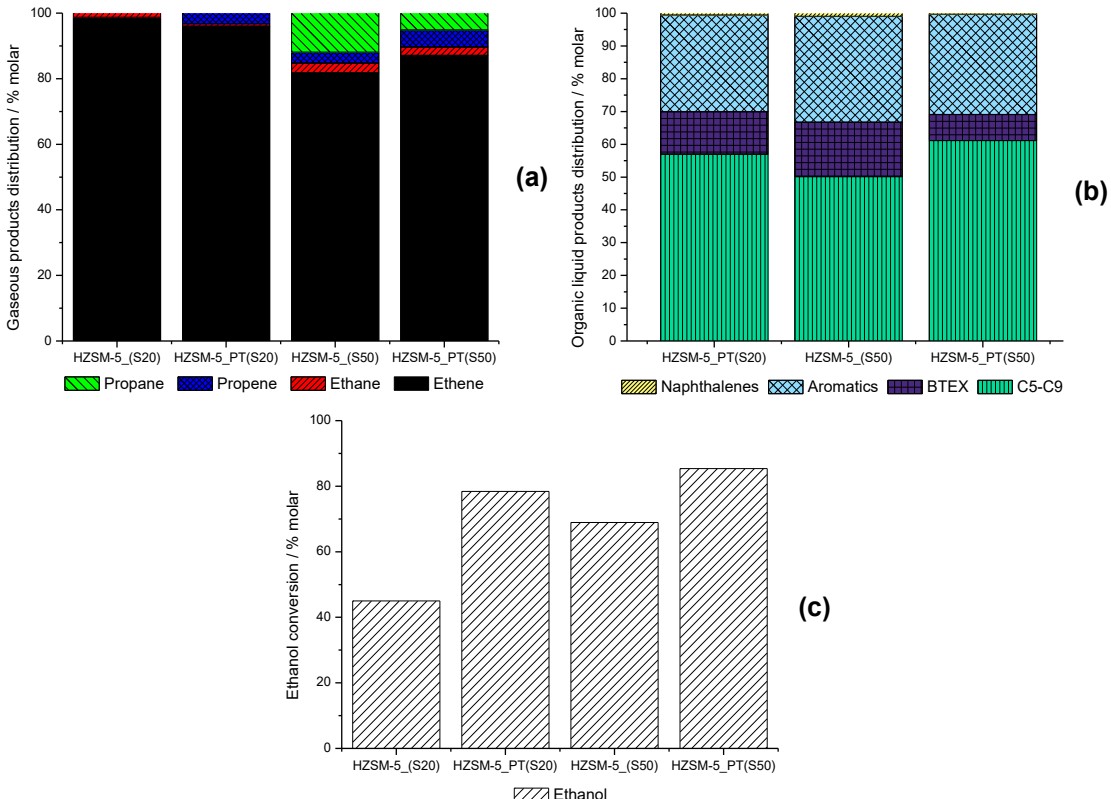

**Figure 3.** Product class distribution of the (**a**) gaseous fraction and (**b**) organic liquid fraction, and (**c**) ethanol conversion, for 360 °C, 15 bar, ~38 h$^{-1}$, and 3 h TOS.

### 2.3. Development of Regression Equation

After performing all the experiments planned by the full factorial central composite design, the multiple linear regression analysis was applied to the set of experimental data obtained. The best model for BTEX experimental response was the quadratic one and the 2FI model was the most suitable for the C2= experimental response. The analysis of variance (ANOVA) was applied to calculate the significance and fitness of the regression models for each experimental response evaluated, further, the significance of the individual terms and their interactions were estimated as well. The regression equations for the experimental responses, variables and its interactions are shown in Equations (1) and (2), where the coded terms A, B, and C are temperature, pressure, and WHSV, respectively.

$$(BTEX + 0.45)^{0.99} = 12.59 + 11.11A - 0.84B - 4.19C - 0.052AB - 4.55AC + 0.18BC \\ + 2.37A^2 + 0.15B^2 + 6.94C^2 \tag{1}$$

$$(C2 =) = 62.64 - 28.43A + 1.42B + 10.14C - 2.29AB + 8.16AC - 0.81BC \tag{2}$$

A *p*-value (probability of error value) less than 0.05 indicates that the parameter is statistically significant for the 95% confidence level. Therefore, according to the data displayed in Tables 2 and 3, it can be seen that for the BTEX and C2= concentrations the models were significant to predict the responses values (*p*-value < 0.0001), and the variation of data around the fitted model (lack of fit) was not significant (*p*-value > 0.05). The two runs related to the axial points for the WHSV variable (13 and 14) showed an outlier behavior and had a negative impact on the data analysis. Therefore, aiming at improving the quality of the multiple linear regression analysis, an outlier filtering was necessary, and these two particular point were removed from the data set. Additionally, by means of the Box Cox Plot tool, the software recommended a power transformation for the BTEX experimental response ($\lambda = 0.99$ and k = 0.45) in order to make the raw data adequate for ANOVA [29,30].

Table 3 show the complete ANOVA for the BTEX regression equation. The best fitting was achieved for the BTEX model, the standard deviation, expressed as a percentage of the mean (C.V. 8.25%), showed a reasonable value, indicating a good degree of precision of the experimental values. According to the BTEX model, the most significant variables and interactions to the BTEX production were (A) temperature, (C) WHSV, AC, $A^2$, and $C^2$, the variable (B) pressure and its interactions were not statistically significant. The evaluation of the coefficients of determination (R-squared) showed that the experimental $R^2$ value was reasonably close to the adjusted $R^2$ and the predicted $R^2$, indicating a good agreement between the predicted and the experimental values (Figure 4).

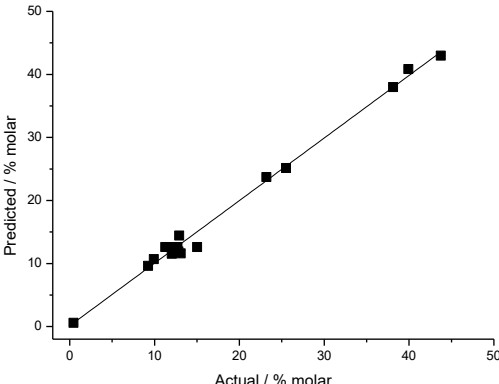

**Figure 4.** Predicted versus actual values for BETX experimental response.

Table 2 show the complete ANOVA for the C2= regression equation. As can be seen, the C.V.% value calculated to the C2= model was 12.24%, in this case, still indicating a reasonable degree of precision, but greater than the BTEX value. According to the C2= model, the most significant variables and interactions to the ethylene production were (A) temperature, (C) WHSV, and AC, the variable (B) pressure and its interactions were not statistically significant in both models, indicating that the chosen range the pressure was not ideal. The evaluation of the coefficients of determination (R-squared) showed that the experimental $R^2$ value was reasonably close to the adjusted $R^2$ and the predicted $R^2$, indicating a good agreement between the predict and the experimental values (Figure 5).

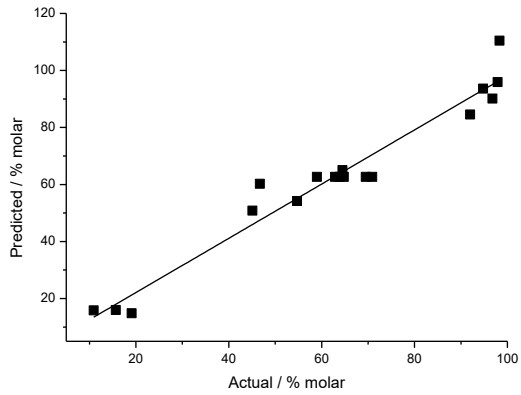

**Figure 5.** Predicted versus actual values for C2= experimental response.

**Table 2.** Analysis of variance for the C2= experimental response.

| Source | Sum of Squares | Df [c] | Mean Square | F Value | *p*-Value Prob > F | Significance |
|---|---|---|---|---|---|---|
| Model | 12470.23 | 6 | 2078.37 | 35.33 | <0.0001 | Highly significant |
| A-T | 11040.27 | 1 | 11040.27 | 187.68 | <0.0001 | Highly significant |
| B-P | 27.66 | 1 | 27.66 | 0.47 | 0.5071 | Not significant |
| C-WHSV | 822.15 | 1 | 822.15 | 13.98 | 0.0033 | Significant |
| AB | 41.86 | 1 | 41.86 | 0.71 | 0.4169 | Not significant |
| AC | 533.01 | 1 | 533.01 | 9.06 | 0.0119 | Significant |
| BC | 5.28 | 1 | 5.28 | 0.09 | 0.7700 | Not significant |
| Residual | 647.07 | 11 | 58.82 | | | |
| Lack of Fit | 550.52 | 6 | 91.75 | 4.75 | 0.0541 | Not significant |
| Pure Error | 96.55 | 5 | 19.31 | | | |
| Std. Dev. [a] | 7.67 | | R-squared | 0.9507 | | |
| Mean | 62.64 | | Adj R-squared | 0.9238 | | |
| C.V.% [b] | 12.24 | | Pred R-squared | 0.8001 | | |
| PRESS | 2622.72 | | Adeq Precision [d] | 19.9952 | | |

($p < 0.0001$) Highly significant; ($0.0001 < p < 0.05$) Significant; ($p > 0.05$) Not significant

[a] Standard of deviation; [b] Coefficient of variation; [c] Degrees of freedom; [d] Adequate precision.

**Table 3.** Analysis of variance for the BTEX experimental response.

| Source | Sum of Squares | Df [c] | Mean Square | F Value | *p*-Value Prob > F | Significance |
|---|---|---|---|---|---|---|
| Model | 2358.78 | 9 | 262.09 | 124.60 | <0.0001 | Highly significant |
| A-T | 1685.52 | 1 | 1685.52 | 801.30 | <0.0001 | Highly significant |
| B-P | 9.67 | 1 | 9.67 | 4.60 | 0.0644 | Not significant |
| C-WHSV | 140.53 | 1 | 140.53 | 66.81 | <0.0001 | Highly significant |
| AB | 0.02 | 1 | 0.02 | 0.01 | 0.9223 | Not significant |
| AC | 165.88 | 1 | 165.88 | 78.86 | <0.0001 | Highly significant |
| BC | 0.25 | 1 | 0.25 | 0.12 | 0.7385 | Not significant |
| $A^2$ | 67.29 | 1 | 67.29 | 31.99 | 0.0005 | Significant |
| $B^2$ | 0.26 | 1 | 0.26 | 0.12 | 0.7328 | Not significant |
| $C^2$ | 182.46 | 1 | 182.46 | 86.74 | <0.0001 | Highly significant |
| Residual | 16.83 | 8 | 2.10 | | | |
| Lack of Fit | 8.20 | 3 | 2.73 | 1.58 | 0.3041 | Not significant |
| Pure error | 8.63 | 5 | 1.73 | | | |
| Std. Dev. [a] | 1.45 | | R-squared | 0.9929 | | |
| Mean | 17.59 | | Adj R-squared | 0.9849 | | |
| C.V.% [b] | 8.25 | | Pred R-squared | 0.9432 | | |
| PRESS | 134.87 | | Adeq Precision [d] | 39.1954 | | |

($p < 0.0001$) Highly significant; ($0.0001 < p < 0.05$) Significant; ($p > 0.05$) Not significant

[a] Standard of deviation; [b] Coefficient of variation; [c] Degrees of freedom; [d] Adequate precision.

### 2.4. Reaction Parameter Study

Figure 6 shows the 3D surface plot and interaction plot for the interaction effect between (B) pressure and (C) WHSV on BTEX molar concentration. When the (A) temperature was fixed at 375 °C it can be seen that, regardless of (B) pressure, higher BTEX concentration was achieved when the (C) WHSV value was at the low level (15 h$^{-1}$). On the other side, the (B) pressure range chosen had no effect on the BTEX concentration, and there was no interaction effect between (B) pressure and (C) WHSV. Lower WHSV values imply in higher contact time, in this case, the light olefins firstly formed tend to behave as reaction intermediates, leading to increasing the paraffin and aromatics yields [10]. Jan and Resende [31] studied the liquid hydrocarbon production via ethylene oligomerization over Ni-Hβ zeolites and, according to the authors, increasing pressure (35 bar to 65 bar) and applying moderate space velocity values (2 h$^{-1}$) promote a higher ethylene conversion and liquid hydrocarbons yield.

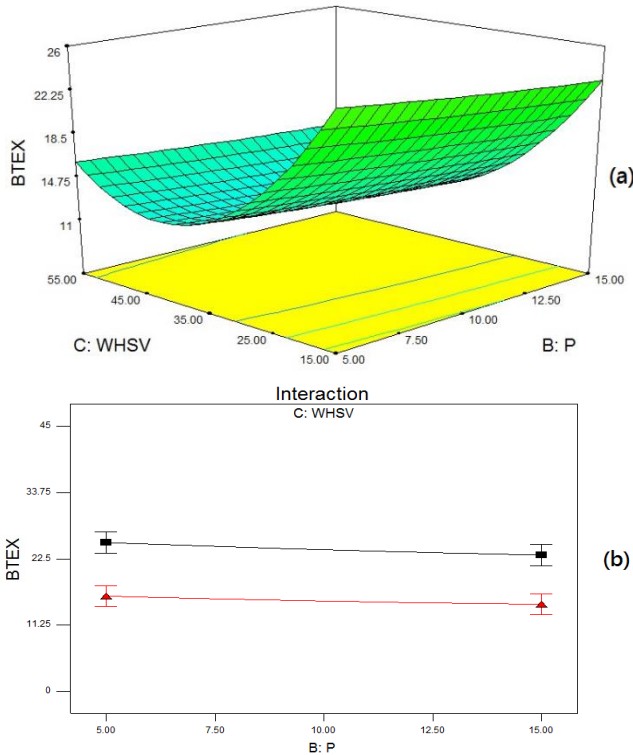

**Figure 6.** Interaction effect study between (B) pressure and (C) WHSV, under BETX (mol%) production. (**a**) 3D surface plot and (**b**) interaction plot.

Similar behavior was found on n-butanol to hydrocarbons, butylenes, and aromatics conversion over acid ZSM-5 zeolites. Palla et al. [32] observed that when WHSV decreased from $14.96\,h^{-1}$ to $0.75\,h^{-1}$, n-butanol conversion increased from 13% to about 72% at 200 °C, and nearly complete conversion was observed at 250 °C. Furthermore, the selectivity for $C_5$–$C_{12}$ hydrocarbons and aromatics also increased at low WHSV. On the other hand, the selectivity to $C_3$–$C_4$ hydrocarbons (intermediates) decreased.

Figure 7 shows the 3D surface plot and interaction plot for the interaction effect between (A) temperature and (B) pressure on BTEX molar concentration. When the (C) WHSV was fixed at $35\,h^{-1}$ it can be seen that, regardless of (B) pressure, higher BTEX concentration was achieved when the (A) temperature value was at the high level (450 °C). On the other side, the (B) pressure range tested had no effect on the BTEX concentration, and there was no significant interaction effect between (B) pressure and (A) temperature. The (A) temperature influence on products distribution (gaseous and liquid) can be better understood in Figure 8. When (B) pressure and (C) WHSV were fixed at 10 bar and $35\,h^{-1}$, respectively, it was observed that, at 249 °C, there was high selectivity for ethylene (>98 mol%). When the temperature increased to 375 °C, besides ethylene production (59 mol%), there was the production of $C_2$–$C_9$ olefins, paraffins, and aromatic compounds. At temperatures higher than 375 °C it was observed that the oligomerization of ethylene was favored, and the formation of heavier olefins and aromatic compounds increased. However, at 501 °C the formation of $C_5$–$C_9$ compounds (non-aromatic) decreased and the formation of aromatics and propane increased. At that temperature, is was observed that compounds such as $H_2$, CO, and $CH_4$ were formed, probably due to cracking of the $C_5$–$C_9$ compounds.

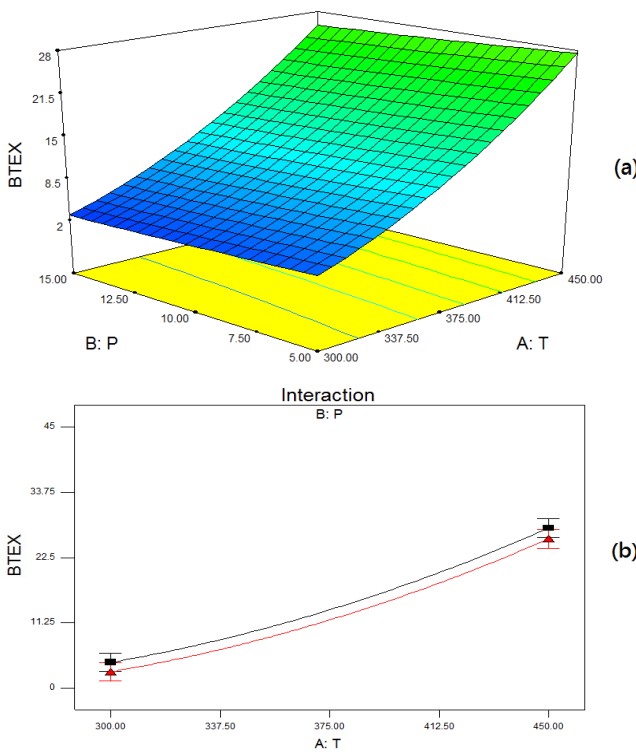

**Figure 7.** Interaction effect study between (B) pressure and (A) temperature, under BETX (mol%) production. (**a**) 3D surface plot and (**b**) interaction plot.

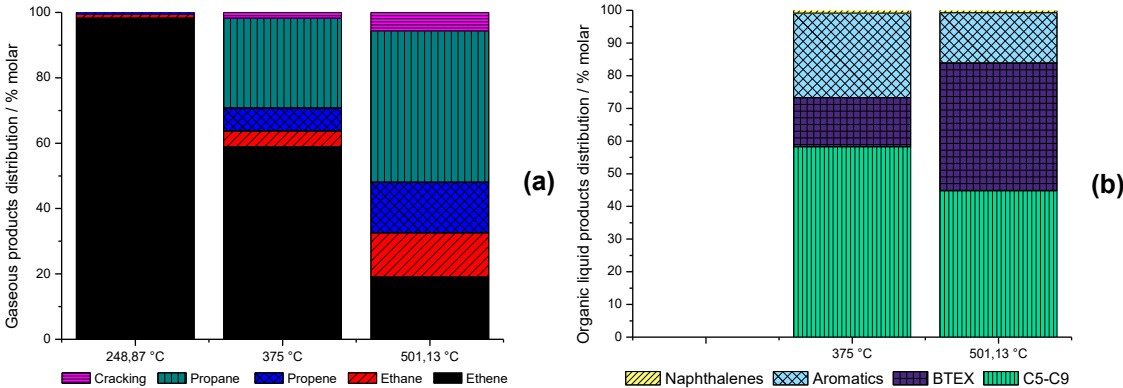

**Figure 8.** Product class distribution of the (**a**) gaseous fraction and (**b**) organic liquid fraction for 10 bar, 35 h$^{-1}$, 3 h TOS, and temperature from 249 °C to 501 °C.

Figure 9 shows the 3D surface plot and interaction plot for the interaction effect between (A) temperature and (C) WHSV on BTEX molar concentration. When the (B) pressure was fixed at 10 bar it can be seen that, regardless of (C) WHSV, higher BTEX concentration was achieved when the (A) temperature value was at the high level (450 °C). According to the ANOVA the interaction between (A) temperature and (c) WHSV was highly significant for the model (*p*-value < 0.0001), from Figure 9 it can be seen that when the temperature was at the low level (300 °C) there was no difference in BTEX production for both high and low levels of WHSV. On the other hand, when the temperature increased from 300 °C to 450 °C the WHSV low level showed to be responsible for higher BTEX production. Among all runs, the highest BTEX production (45 mol%) was reached for 450 °C, 5 bar, and 15 h$^{-1}$, for 99.5% ethanol conversion (3 h TOS).

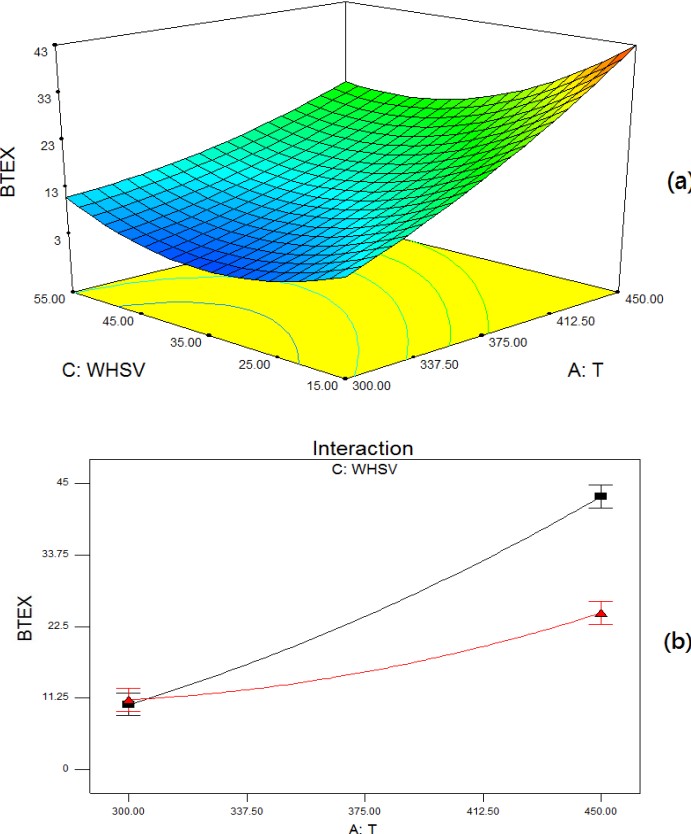

**Figure 9.** Interaction effect study between (C) WHSV and (A) temperature, under BETX (mol%) production. (**a**) 3D surface plot and (**b**) interaction plot.

### 2.5. Optimization of BTEX and Ethylene Production

The optimization step was performed aiming at finding the best combination of temperature, pressure, and space velocity, at which the maximum of BTEX and the minimum of ethylene concentrations could be achieved. The optimal values generated by the software were 450 °C, 20 bar, and 5 h$^{-1}$. The predicted BTEX and ethylene concentrations were 54.8 mol% and 7.5 mol%, respectively. In order to check the predictive ability of the regression model, three independent replicates were carried out, the experimental values obtained for BTEX and ethylene concentrations were 49.3 ± 2.3% and 6.9 ± 2.4%, respectively. Considering the standard deviation expressed as the percentage of the mean for both models (BTEX = 8.25% and ethylene = 12.24%) the experimental values obtained showed reasonable accuracy on predicting the optimal point on BTEX and ethylene concentrations.

### 2.6. Time-on-Stream Evaluation of HZSM-5_(S50)

The stability of the HZSM-5_(S50) catalyst was also investigated (Figure 10). Complete ethanol conversion was observed for the entire 10 h TOS evaluation at 375 °C, 10 bar, and 70 h$^{-1}$. On the other hand, both gaseous and organic liquid products' distribution changed with increasing TOS. As can be seen in Figure 10a, ethene selectivity increased with longer TOS and heavier products' selectivity (propane and propene) decreased consequently.

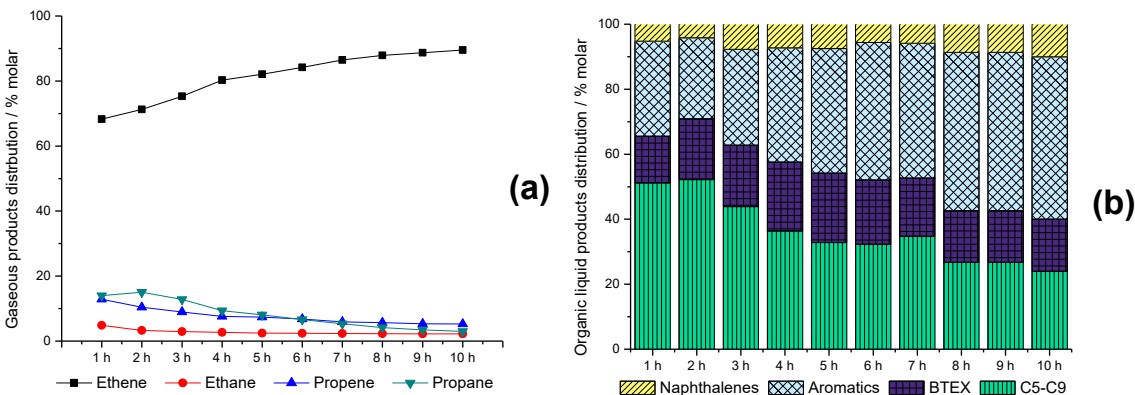

**Figure 10.** Product class distribution of the (**a**) gaseous fraction and (**b**) organic liquid fraction, for 375 °C, 10 bar, ~70 $h^{-1}$, and 10 h TOS catalyst stability test.

The organic liquid fraction showed a similar behavior, as can be seen in Figure 10b, and the production of $C_5$–$C_9$ olefins and paraffins slightly decreased with TOS increasing as well. Both behaviors were already expected, and can be associated with a slight deactivation of the catalyst by coke deposition.

## 3. Materials and Methods

The HZSM-5 zeolites gently provided by PROCAT (a scale-up concept for catalyst production with a complete infrastructure to manufacture any type of heterogeneous catalysts, located in Rio de Janeiro, Brazil) [33] have framework SAR values equal to 23 and 53. In order to increase the zeolites' pore sizes, two hierarchical zeolites were synthesized by means of a desilication technique. Thus, a screening of catalyst (all four zeolites) and variables focused on reaching the adequate conditions for the optimization study was made. The desilication procedure was performed based on Xin's group's manuscript [18] (previously cited), the zeolite was mixed in a 0.2 mol/L NaOH solution (3.3 g of zeolite to 100 mL NaOH solution) and vigorously stirred in a propylene flask at 65 °C for 30 min. Afterward, the suspension was filtered and washed three times with 50 mL of deionized water. In order to convert the alkali-treated zeolites into H-form, three consecutive ion exchanges using a 0.1 mol/L $NH_4NO_3$ solution was performed (1 g of zeolite to 30 mL $NH_4NO_3$ solution). The ammonium zeolite was calcined in static air at 550 °C for 3h (heating rate 1 °C/min). HZSM-5_PT(S20) and HZSM-5_PT(S50) nomenclature refer to the hierarchical acid zeolites with SAR close to 20 and 50, respectively. Throughout the manuscript, the parent acid zeolites with SAR 20 and 50 will be named as HZSM-5_(S20) and HZSM-5_(S50), respectively.

### 3.1. Catalyst Characterization

X-ray diffraction (XRD) patterns of powder crystalline samples were collected at ambient temperature on a Rigaku MiniFlex II (The Woodlands, TX, USA) diffractometer using Cu αK radiation (wavelength λ = 0.15488 nm), the spectra scanning range was from 5° to 40°. The XRD crystallinity was obtained as described by Xin et al. [18], the ratio of the sum of the three most intense reflections in the 2θ range from 20° to 25°, and the corresponding sum for the parent zeolites' peak intensities was calculated (the parent zeolites peak intensities were considered to be 100% crystallinity). Nitrogen adsorption-desorption isotherms were measured at −196 °C on a Micromeritcs Tristar II (Norcross, GA, USA), before the analysis each sample was degassed at 300 °C for 12 h. The total surface area was calculated according to the Brunauer–Emmet–Teller method (BET), the micropore volume and the external surface area (meso-area) were calculated using the t-plot method, and the pore size distribution was calculated by means of the Barret–Joyner–Halenda method (BJH). The HZSM-5 zeolites' SAR (calculated as the $SiO_2/Al_2O_3$ molar ratio) were determined by X-ray fluorescence (XRF) on a Bruker S2 Ranger spectrometer.

### 3.2. Catalytic Evaluation

The ethanol conversion reaction was carried out in an automated continuous flow catalytic evaluation unit (Microactivity Effi reactor—PID Eng&Tech—Alcobendas, Madrid, Spain) [34] equipped with a fixed-bed down-flow Hastelloy C273 tube reactor (13.1 mm internal diameter). The pre-established catalyst amount (60–100 mesh) was mixed with silicon carbide (200 mesh) and placed on a quartz wool bed in the middle of the reactor. Before the reaction began the catalyst was thermally pre-treated under $N_2$ flow (20 mL min$^{-1}$) for 1 h at 350 °C, the same $N_2$ flow was maintained throughout the reaction. Once the reaction conditions were reached, the anhydrous ethanol was fed into the reactor and kept at the desired flow. The condensable reaction products were cooling down and collected by an inline Peltier separator, the gaseous products were analyzed by an online gas chromatograph (GC) equipped with a barrier ionization detector (BID) and a Carboxen-1010 PLOT capillary column (30 m × 0.32 mm × 15μm, Supelco/Sigma Aldrich—St. Louis, MO, USA). The organic liquid products (if any) were analyzed by a GC equipped with a flame ionization detector (FID) and a DB-5HT capillary column (15 m × 0.25 mm × 0.1 μm, J and W Scientific/Agilent—Santa Clara, CA, USA) and a GC equipped with a mass spectrometer detector and a HP-5MS column (30 m × 0.25 mm × 0.1 μm, J and W Scientific/Agilent—Santa Clara, CA, USA), the aqueous fraction was analyzed by a high performance liquid chromatograph equipped with a refractive index detector (RID) and a Aminex HPX-87H column (300 mm × 7.8 mm, BioRad—Hercules, CA, USA).

### 3.3. Experiment Design and Statistical Study

The experimental design was provided by Design-Expert software 7.0.0 (Stat-Ease, Inc., Minneapolis, MN, USA). The reaction parameters: temperature (A), pressure (B), and space velocity (C) were chosen as independent variables (factors), and BTEX and C2= (ethylene) molar concentrations (mol%) were chosen as experimental responses. A five-level-three-factor central composite design was employed, where 20 experiments were required (eight factorials, six axial, and six center point replicates), and based on the initial screening of the catalyst only the HZSM-5_(S50) zeolite was used as the catalyst for the statistical study.

With the purpose of minimizing errors, the experiments were run in random order, Table 4 shows the range and levels of the independent variables and Table 5 shows the complete matrix with the 20 runs and the experimental responses values. For a rotatable design, the alpha values were fixed at −1.682 (−$\alpha$) and +1.682 (−$\alpha$). The polynomial equation model responsible for predicting the optimal point for the experimental responses based on interaction among variables is expressed by Equation (3), where Y is the predicted response, β$o$, β$j$, β$ij$, and β$jj$ are the constant coefficients, x$i$ and x$j$ are the coded independent variables, and $\varepsilon$ is the random error:

$$Y = \beta_o + \sum_{j=1}^{k} \beta_j x_j + \sum \sum_{i<j} \beta_{ij} x_i x_j + \sum_{j=1}^{k} \beta_{jj} x_j^2 + \varepsilon \tag{3}$$

**Table 4.** Experimental range and coded values of independent variables for the ethanol conversion into hydrocarbons process.

| Factor | Coding | Units | Levels | | | | |
|---|---|---|---|---|---|---|---|
| | | | −1.682 | −1 | 0 | 1 | 1.682 |
| Temperature | A | °C | 248.9 | 300.0 | 375.0 | 450.0 | 501.1 |
| Pressure | B | Bar | 1.6 | 5.0 | 10.0 | 15.0 | 18.4 |
| WHSV | C | h$^{-1}$ | 1.4 | 15.0 | 35.0 | 55.0 | 68.6 |

**Table 5.** Full factorial central composite design matrix and experimental data of the ethanol conversion into hydrocarbons.

| Point | Std [a] | T (°C) | P (bar) | WHSV (h$^{-1}$) | BTEX (mol%) | C2 = (mol%) |
|-------|---------|--------|---------|------------------|-------------|-------------|
| Factorial | 1 | 300.0(−1) | 5.0(−1) | 15.0(−1) | 11.9 | 92.0 |
| Factorial | 2 | 450.0(+1) | 5.0(−1) | 15.0(−1) | 45.0 | 15.7 |
| Factorial | 3 | 300.0(−1) | 15.0(+1) | 15.0(−1) | 9.0 | 94.8 |
| Factorial | 4 | 450.0(+1) | 15.0(+1) | 15.0(−1) | 41.0 | 10.9 |
| Factorial | 5 | 300.0(−1) | 5.0(−1) | 55.0(+1) | 12.7 | 96.8 |
| Factorial | 6 | 450.0(+1) | 5.0(−1) | 55.0(+1) | 25.9 | 54.7 |
| Factorial | 7 | 300.0(−1) | 15.0(+1) | 55.0(+1) | 9.7 | 97.9 |
| Factorial | 8 | 450.0(+1) | 15.0(+1) | 55.0(+1) | 23.5 | 45.1 |
| Axial | 9 | 248.9(−1.682) | 10.0(0) | 35.0(0) | 0.0 | 98.3 |
| Axial | 10 | 501.1(+1.682) | 10.0(0) | 35.0(0) | 39.1 | 19.1 |
| Axial | 11 | 375.0(0) | 1.6(−1.682) | 35.0(0) | 12.8 | 46.7 |
| Axial | 12 | 375.0(0) | 18.4(1.682) | 35.0(0) | 13.0 | 64.5 |
| Axial | 13 | 375.0(0) | 10.0(0) | 1.4(−1.682) | 40.0 | 1.6 |
| Axial | 14 | 375.0(0) | 10.0(0) | 68.6(1.682) | 20.0 | 57.0 |
| Center | 15 | 375.0(0) | 10.0(0) | 35.0(0) | 15.0 | 59.0 |
| Center | 16 | 375.0(0) | 10.0(0) | 35.0(0) | 12.6 | 64.8 |
| Center | 17 | 375.0(0) | 10.0(0) | 35.0(0) | 12.5 | 63.9 |
| Center | 18 | 375.0(0) | 10.0(0) | 35.0(0) | 11.6 | 62.9 |
| Center | 19 | 375.0(0) | 10.0(0) | 35.0(0) | 12.0 | 70.9 |
| Center | 20 | 375.0(0) | 10.0(0) | 35.0(0) | 11.1 | 69.5 |

[a] The experiment order created by CCD.

## 4. Conclusions

The alkali treatment of the ZSM-5 zeolites generated mesoporosity with good maintenance of crystallinity. The regression models generated explained the experimental data with reasonable accuracy, all tested zeolites were active on ethanol conversion, however, the HZSM-5_(50) exhibited the most diversified range of products (organic liquid and gaseous fractions) and the HZSM-5_PT(50) was the most active catalyst with 85.4% of ethanol conversion at 360 °C, 15 bar, ~38 h$^{-1}$, and 3 h TOS. The modeling results showed that the HZSM-5_(50) zeolite optimized conditions for ethanol conversion were 450 °C, 20 bar, and 5 h$^{-1}$, and considering the standard deviation expressed as a percentage of the mean for both models, the predictive ability of the regression model showed reasonable values.

**Author Contributions:** Conceptualization, J.F.S.d.C.F., M.M.P., D.A.G.A., J.M.A.R.d.A., E.F.S.-A. and P.N.R.; methodology, J.F.S.d.C.F. and P.N.R.; validation, J.F.S.d.C.F., D.A.G.A. and P.N.R.; formal analysis, J.F.S.d.C.F. and P.N.R.; investigation, J.F.S.d.C.F., D.A.G.A. and P.N.R.; resources, M.M.P., D.A.G.A. and E.F.S.-A.; data curation, J.F.S.d.C.F. and P.N.R., writing-original draft preparation, J.F.S.d.C.F.; writing-review and editing, J.F.S.d.C.F., M.M.P., D.A.G.A., J.M.A.R.d.A., E.F.S.-A. and P.N.R.; visualization, J.F.S.d.C.F., D.A.G.A., E.F.S.-A. and P.N.R.; supervision, M.M.P., D.A.G.A. and E.F.S.-A.; project administration, J.F.S.d.C.F.

**Funding:** This research received no external funding.

**Acknowledgments:** The authors would like to thank Coordenação de Aperfeiçoamento de Pessoal de Nível Superior (CAPES) for the fellowships, and Laboratório de Tecnologias do Hidrogênio (Labtech) for XRD and XRF characterizations.

**Conflicts of Interest:** The authors declare no conflict of interest.

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
