# Peer review of "Application of Response Surface Methodology for Ethanol Conversion into Hydrocarbons Using ZSM-5 Zeolites"

_catalysts, doi:10.3390/catal9070617_

Round 1

Reviewer 1 Report

The authors should report the catalyst durability (deactivation) of the used ZSM-5 catalysts.

Reviewer 2 Report

This paper deals on the performance of two ZSM-5 zeolites of different si/Al ratio in the conversion of ethanol to other products. The influence of alkaline treatment in the porous structure has been explored. Finally, an statistical analysis has been performed to obtain the optimal conditions and properties of the material studied. The paper is well written and easy to read, and I only have a few curiosities before being considered for publication:

The first mentioned to the four zeolites tested in this work in page 3, line 102, does not allow to discern from Figure 1 or table 1 the influence of their differences in the results that are explained below. Otherwise, it is very difficult to understand the purpose and the method of silicon removing from the zeolite structure, as explained in lines 112-126.

 In the materials and methods section in page 10, lines 272-285, the authors only say that the zeolites were provided by PROCAT. It is recommended that the authors add the full affiliation of PROCAT so that the reader knows at once what kind of institution is. Are the zeolites commercial or made in the lab facilities of PROCAT? Please clarify.

There are two different HZSM-5 zeolites whose only apparent difference is the Si/Al ratio is 20 or 50; and only the high Si/Al ratio zeolite allows for mesopores being generated by the Si dissolution in NaOH treatment, while the low Si/Al ratio does not. The untreated high Si/Al ratio zeolite is showing a higher variability of the products obtained from the reaction, while the treated zeolite leads to the highest ethanol conversion. Nevertheless, only the parent zeolite’s results are studied by the statistical analysis to obtain the regression equation and the optimal conditions. I think the statistical model is applied for the best catalyst as reference for this work and the extension of the results to other catalytic systems.

English language is generally correct but a careful revision is recommended too.

Reviewer 3 Report

The work presented by the authors describe two stages:

- A first one, where two samples were desilicated in alkali medium an tested in ethanol reaction.

- The second one, where one sample where tested in ethanol reaction over 3 opeartion variables attending to a experimental design developed in order to obtain an expresion for BTEX and ethylene predition.

Whit respect to the first one. Discussion should be revised in term of conversion intead of activity (which it is not calculated and referenced at all). Additionally, the analysis of reaction distribution products should be compared in similar conversion terms in order to evade kinetical artefacts,  due i.e. to limitation of reactant concentration at high conversion states or bysides reaction interference. 

Additionally, authors should revise or explain why or how is SAR value obtained (p.4 line 128-129 indicate XRD, but probably it should be XRF). In experimental section should be included how the crystallinity were calculed and why BJH adsorption branch is selected (in spite of desorption one).

Whit respect to the second one. Discussion is not supported for all the four catalyst, so it should be revised in "abstract", neither SRS methodology with CCD, but an extension of a previous work, This experimental design is not discussed and the variables selected have a scarce catalytic correlation (neither the boundary conditions nor the variable selection). It should be expected SAR, mesoporous, microporous instead of T, P and contact time  attending to the title of the work. So, there is little interest in these expresions for ethylene and BTEX prediction.

In this form, I can not suggest the publication of this work.

Round 2

Reviewer 3 Report

Authors have been answer to the previous suggestions